# StructLM: Towards Building Generalist Models for Structured Knowledge Grounding

**Alex Zhuang**[1*]   **Ge Zhang**[1,2,7*],
**Tianyu Zheng**[2],   **Xinrun Du**[2],   **Junjie Wang**[3,4],   **Weiming Ren**[1,7],
**Stephen W. Huang**[6],   **Jie Fu**[2,4†]   **Xiang Yue**[2,5],   **Wenhu Chen**[1,2,7†]
[1]University of Waterloo, [2]Multimodal Art Projection Research Community,
[3]Waseda University, [4]HKUST, [5]Ohio State University, [6]harmony.ai, [7]Vector Institute
https://tiger-ai-lab.github.io/StructLM/

## Abstract

Structured data sources, such as tables, graphs, and databases, are ubiquitous knowledge sources. Despite the demonstrated capabilities of large language models (LLMs) on plain text, their proficiency in interpreting and utilizing structured data remains limited. Our investigation reveals a notable deficiency in LLMs' ability to process structured data, e.g., Chat-GPT lags behind state-of-the-art (SoTA) model by an average of 35%. To augment the Structured Knowledge Grounding (SKG) capabilities in LLMs, we have developed a comprehensive instruction tuning dataset comprising 1.1 million examples. Utilizing this dataset, we train a series of models, referred to as StructLM, based on Mistral and the CodeLlama model family, ranging from 7B to 34B parameters. Our StructLM series surpasses task-specific models (Xie et al., 2022) on 16 out of 18 evaluated datasets and establishes new SoTA performance on 8 SKG tasks. Furthermore, StructLM demonstrates strong generalization across 6 novel held-out SKG tasks, outperforming TableLlama by an average of 35% and Flan-UL2 20B by an average of 10%. Contrary to expectations, we observe that scaling model size offers marginal benefits, with StructLM-34B showing only slight improvements over StructLM-7B. This suggests that structured knowledge grounding is still a challenging task and requires more innovative design to push to a new level. We release the model weights and training dataset to the community, along with relevant code on Github.

## 1 Introduction

Traditionally, users need to write programs to interface with structured data like tables, databases, knowledge graphs, etc. This often requires that they master a the domain-specific language like SQL, SPARQL, etc. or develop domain specific skills for that structured type. Recently, researchers have explored the possibility of automating the interface with natural language to enable potential use cases in question-answering (Pasupat & Liang, 2015; Zhong et al., 2017; Nan et al., 2022), summarization (Parikh et al., 2020; Nan et al., 2021; Bao et al., 2018), and fact verification (Aly et al., 2021; Chen et al., 2019; Gupta et al., 2020b), among others, all grounded to a structured knowledge source. This effort can lower the barrier for end users to access a massive amount of structured data.

Previous work (Yu et al., 2020; Liu et al., 2021; Xie et al., 2022; Zhang et al., 2023) has been mostly focused on building task-specific models for different tasks with rather limited generalization ability. Building a generalist structure knowledge grounding (SKG) system across a wide range of tasks proves to be challenging. This is mainly due to the heterogeneity of data format and use cases. We evaluated GPT-3.5-Turbo (Jiang et al., 2023) on 18 SKG tasks and observed that its performance is on average 35% lower than the SoTA specialized models. It shows that the LLM's ability on SKG is heavily overlooked during pre-training.

In this paper, we explore the possibility of building a generalist model based on LLMs that can ground on diverse types of structure and unstructured knowledge to interface with humans. Specifically, we construct a large data set of over a million instruction-following examples, a majority of which is SKG data, along with additional general instruction-following data, which we find improves generalizability. We fine-tune models at three scales: 7B, 13B, and 34B, based on Mistral and the CodeLlama family of code foundation models. When compared to USKG, we find that our models surpass these single-task models on 16 of 18 tasks. `StructLM` achieves SoTA on 8 of 18 evaluated tasks, beating ChatGPT by a huge margin. Morever, we show that compared with TableLlama, a recent open language model finetuned for tabular inference tasks, our held-out performance is vastly superior. Compared with Flan-UL2-20B, our models also generalize better overall on the held out tasks by as much as 10% on average.

We study the performance of `StructLM`, namely whether the model experiences cross-task generalization benefits from the dataset mixture, and find that our multi-task model performs significantly better overall than single-task models of the exact same parameter scale. We also study the effect of different pretraining data on our finetuned performance to determine whether special pretraining regimes, such as code or math, contribute to effective SKG reasoning ability. We find that code pretraining is the most effective. We perform additional ablations to confirm our results and support our claims. Our contributions are:

- We construct a large SKG instruction-tuning dataset with 1.1 million samples. We train and release our models that outperform the previous 3B USKG fine-tuned on individual tasks on a total of 16 of 18 tasks. `StructLM` also achieves SoTA results on 8 of them.

- We show that `StructLM` is able to show strong zero-shot generalization capability on unseen structure knowledge grounding tasks, which was not shown by previous models. `StructLM` outperforms Flan-UL2 20B on 3 of 5 tasks and TableLlama on all 6 tasks. The average absolute improvement over them are 10% and 35%.

- We find that mixing in general-domain instruction-tuning data during finetuning preserves generalization ability, and that code-pretrained base models can improve model performance on the SKG tasks.

## 2 Related Works

### 2.1 Solving SKG tasks

Structured knowledge, such as web tables, knowledge graphs, and databases, have long been the subject of study in knowledge grounding. However, SKG tasks have heterogeneous data formats which have inspired methods that leverage specific training setups to

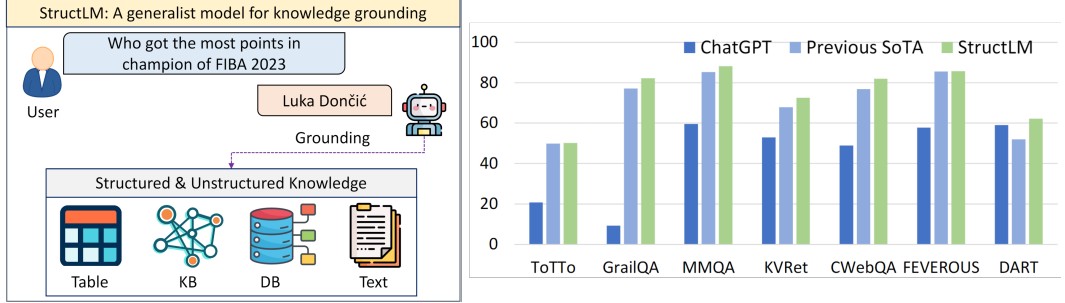

Figure 1: `StructLM` can ground on structured and unstructured knowledge to respond to human queries. The previous SoTA was attained by many different task-specific models like TAPEX (Liu et al., 2021), USKG (Xie et al., 2022), TableLlama (Zhang et al., 2023), BINDER-Codex (Cheng et al., 2022), etc. `StructLM` beats the SoTAs on seven SKG tasks.

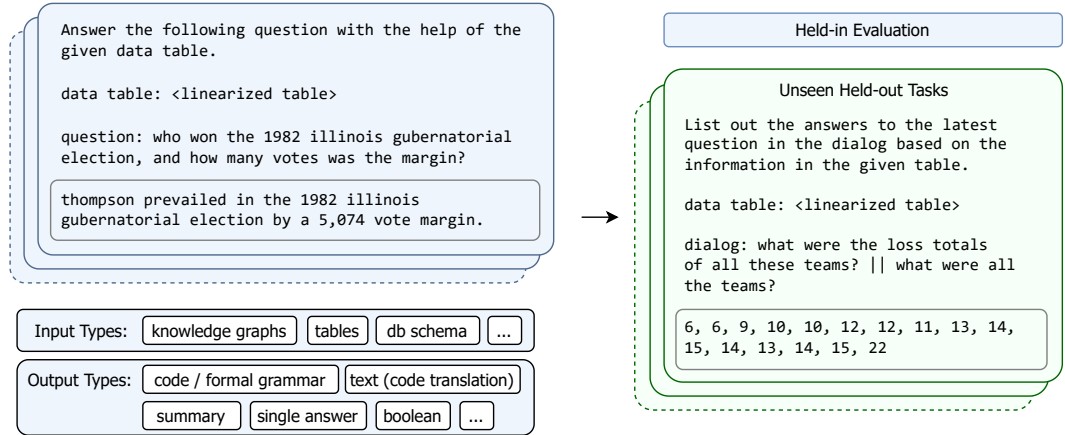

Figure 2: Overview of StructLM. This figure illustrates the prompting structure of StructLM, highlighting its capability to process various forms of structured data beyond linearized data tables, including linearized database schemas and knowledge graphs.

learn those representations. For example, PTab (Liu et al., 2022) and MultiHiertt (Zhao et al., 2022) learn the contextual representation of tabular data by incorporating semantic information through specific training methods or reasoning approaches. RASAT (Qi et al., 2022a) integrates relation-aware self-attention with the Transformer seq2seq architecture and utilizes various relational structures to address SQL problems. TAPEX (Liu et al., 2021) conducts pretraining over tabular/database data with the help of an SQL executor to provide supervision.

More recently, methods have begun to move away from these auxiliary task-specific structures. USKG (Xie et al., 2022) were the first to unify many SKG tasks into a sequence-to-sequence format, allowing them to be aggregated into the same data mixture. However, they were not able to show strong performance improvements to constructing a multi-task mix of SKG data over task-specific tuning methods. StructGPT (Jiang et al., 2023) represents a line of work that uses prompting frameworks on powerful LLMs to solve tasks with more robustness and accuracy. In contrast, our work examines open models and tries to assess their fundamental capabilities. Contemporary to our work, TableLlama (Zhang et al., 2023) has argued that tabular data deserves special attention. Focusing on this domain, their method fine-tunes on several new tabular tasks to improve table understanding, and operates on a longer 8k context length. These improvements can be additive to our work.

## 2.2 LLMs with Instruction Tuning

Instruction-tuning (IT) has been popularized as a method to address the gap between training objectives and user goals in LLMs. This technique involves additional training of LLMs using pairs of instructions and outputs. IT enhances both the controllability and the predictability of the models, aligning them more closely with user expectations. Furthermore, recent studies such as FLAN (Wei et al., 2022), UL2 (Tay et al., 2023a), and Llama2 (Touvron et al., 2023) have shown that IT can improve the performance of downstream tasks through multi-task learning across diverse data types. While FLAN-UL2 trains on a subset of 11 SKG tasks, it also trains on many more unrelated language tasks. In our work, by focusing on SKG data, we hope to provide a focused study that can act as a reference for future work to improve performance on this task type.

## 2.3 Reasoning Capability in LLMs

Reasoning stands as a pivotal skill for LLMs in the development of real-world AI applications which would enable the autonomous completion of many thought-intensive tasks viewed traditionally to require human thinking, like programming or mathematical problem-solving (Li et al., 2022). Recent studies (Li et al., 2022; 2023c; Rozière et al., 2023;

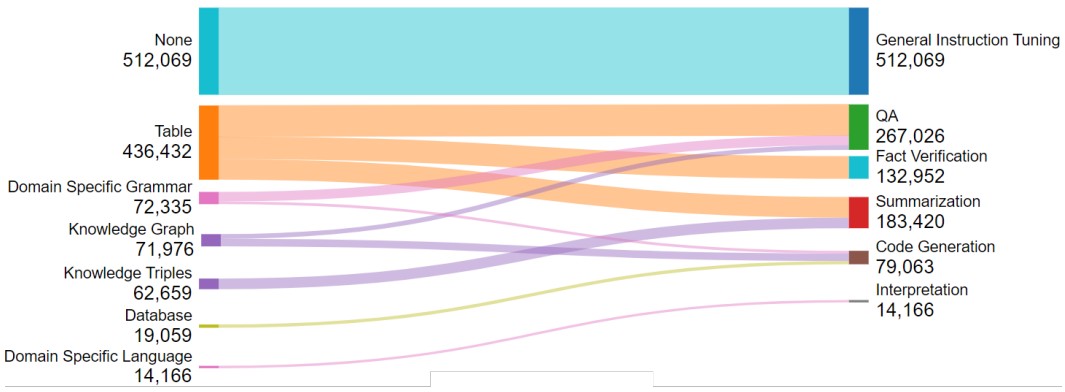

Figure 3: Breakdown of Structured Knowledge Types and Tasks in the Training Data. On the left side, we see a coarse breakdown of the different categories of structured inputs in our dataset. On the right side, we see an overview of the task groups that are represented for those structured knowledge types.

Azerbayev et al., 2023a) indicate that LLMs trained on code and mathematical datasets exhibit profound reasoning skills, and can even achieve performance on par with human levels. For example, CodeLlama (Rozière et al., 2023), a foundation model trained on more programming data, has significantly improved reasoning capabilities across a variety of programming and mathematical benchmarks. Furthermore, Llemma (Azerbayev et al., 2023a) continues to pretrain the CodeLlama model on a mix of scientific papers, math-related web data, and mathematical code. Its results show excellent reasoning capabilities on the MATH benchmark (Hendrycks et al., 2021) and the ability to prove theorems without further fine-tuning. On the fine-tuning side, WizardMath (Luo et al., 2023a), and WizardCoder (Luo et al., 2023c) have shown the effectiveness of instruction tuning on reasoning capabilities, given high quality data.

In this work, we view structured data as a third testbed for a different kind of reasoning within LLMs. We posit that in addition to mathematical or logical reasoning, the ability to recognize and make use of patterns within a structured input indicates that a model has robust representations of relationships in data. These representations may serve as a strong prior for further reasoning downstream.

## 3 Method

### 3.1 Dataset Curation

Motivated by the goal of training a language model generally capable of a wide range of structured data tasks, we select a total of 25 SKG tasks to study. We report results on 18 held-in and 6 held-out tasks, where each held-out task meant to roughly evaluate the generalization capability of a held-in task group. In total, our held-in training dataset contains approximately 700k SKG examples. We describe the held-in dataset groups below.

**Data to Text Generation**. This group of datasets deals with the summarization or interpretation of structured data from tables to knowledge triples to formal languages. Their inclusion is motivated by the idea that useful LMs should be able to make sense of a wide variety of structured information and map it to meaning in natural language. The corresponding held-out dataset for this task group is intended to be WikiTableText.

**Table based Question Answering**. This group of datasets deals specifically with tabular data, optionally combined with text passages. LMs which are able to accurately answer questions and retrieve information from tables can be widely useful as assistants. The corresponding held-out dataset for this task group is SQA.

**Knowledge-grounded Conversations**. This group of tasks evaluates knowledge grounding in-conversation. Humans naturally interface with LMs through chat, and enabling this capability can lower the barrier to accessing the information in stored structured data. These tasks track user intention through provided dialogue and ask the model to provide an answer to the latest question. The held-out dataset for this task group is CoSQL.

**Fact verification**. One common use case for tables is to reference facts. In addition to question answering, the ability to reliably determine if data in a table supports a statement signals the existence of a robust representation of the table's data. The held-out dataset for this task group is InfoTabs.

**SQL or domain-specific languages** SQL is the language most commonly used to interface with structured data today. Understanding how to write SQL also requires understanding of abstractions of tables and how they are linked together. In other domain-specific languages, the MTOP task measures a model's ability to parse a specification and generate an API call, which sees potential in LLM tool use (e.g., (Qin et al., 2023)). The corresponding held-out dataset for this task group is intended to be BIRD (Li et al., 2023b), which further tests SQL generation abilities.

**Mathematical reasoning**. An analysis of tabular data may also require performing quick mathematical computations over their contents. Performance on these datasets tells us how well models can combine both structured knowledge and mathematical reasoning. As there are currently a limited number of datasets that combine mathematical reasoning with SKG, this category includes just TabMWP in the held-in corpus. We set FinQA as a challenging held-out dataset analogue. Not only does it require financial domain knowledge, but it combines tabular information with long text passages, and requires the generation of mathematical code.

**General instruction data**. In addition to the SKG datasets within the held-in dataset mixture, we also included general instruction tuning data without any structured knowledge component, to maintain the instruction-following ability of our model. We use SlimOrca (Lian et al., 2023), which is constructed from cleaned GPT-4 responses to a number of prompts from existing general large-scale instruction-following datasets. We detect no signs of data contamination for our held-out datasets based on our ablation results. We give a detailed overview of all dataset statistics in Table 1.

## 3.2 Instruction Finetuning Approach

To instruction tune our model, each example in our dataset consists of a system prompt, instruction, input, and output. For all SKG data examples, we use the same system prompt. For each dataset, we write 10 instruction variations, which are randomized when constructing the training samples. For SKG data, the input is composed of a combination of a structured knowledge input and accompanying text that could be a question, statement, or anything that would be required to specify the task. The prompt is provided in Figure 6.

## 3.3 Training and Evaluation Details

The base models for StructLM are the CodeLlama-Instruct family of models (Rozière et al., 2023). We finetune all models with a batch size of 512 for 3 epochs on A800 gpus. We train our 7-34B models on 16-64 GPUs using DeepSpeed ZeRO-3 (Rasley et al., 2020) such that the total training time for each model is between 3-5 days. This training setup is largely in line with community conventions, such as the settings used for the WizardLM (Xu et al., 2023), WizardMath (Luo et al., 2023a), and WizardCoder (Luo et al., 2023c) models.

We follow the structured data linearization conventions in USKG (Xie et al., 2022). However, we use a different truncation scheme as described below. During training, we maintain a maximum sequence length of 2048. To preserve as much input context as possible, when truncating we consider the combined token length of the prompt input and output label. We truncate only the structured knowledge portion of the input so that the example becomes at most 2048 tokens long. As shown in the dataset statistics in Table 1, setting the max token length of the examples in our dataset to 2048 allows nearly all examples to fit within the

| Dataset | Overall Length | | Train | | | | Test | | | |
|---------|----------------|---|-------|---|---|---|------|---|---|---|
| | Input (avg) | Output (avg) | Count | Input (max) | Output (max) | # Trunc. | Count | Input (max) | Output (max) | # Trunc. |
| TabMWP | 207.8 | 4.5 | 23059 | 709 | 33 | 0 | 7686 | 703 | 31 | 0 |
| ToTTo | 251.8 | 31.0 | 120761 | 2040 | 155 | 467 | 7700 | 2048 | 119 | 31 |
| GrailQA | 281.0 | 44.1 | 44337 | 884 | 134 | 0 | 6463 | 546 | 123 | 0 |
| SQL2Text | 122.3 | 18.1 | 5600 | 337 | 61 | 0 | 1034 | 245 | 38 | 0 |
| MMQA | 656.2 | 7.7 | 15688 | 2047 | 146 | 234 | 1501 | 2048 | 94 | 11 |
| Spider | 266.6 | 36.0 | 7000 | 1369 | 226 | 0 | 1034 | 453 | 146 | 0 |
| KVRet | 573.4 | 17.1 | 6288 | 1217 | 161 | 0 | 807 | 1147 | 82 | 0 |
| HybridQA | 700.4 | 6.8 | 62682 | 2047 | 91 | 200 | 3466 | 2048 | 79 | 6 |
| SParC | 276.3 | 32.6 | 12059 | 1417 | 226 | 0 | 1625 | 467 | 146 | 0 |
| CompWebQ | 1350.3 | 11.9 | 27639 | 2047 | 321 | 321 | 2816 | 2048 | 256 | 8 |
| TabFact | 660.1 | 4.6 | 92283 | 2045 | 5 | 2 | 12779 | 1687 | 4 | 0 |
| WikiTQ | 831.8 | 5.8 | 11321 | 2028 | 273 | 0 | 4344 | 2048 | 148 | 10 |
| WikiSQL | 689.2 | 7.1 | 56355 | 2047 | 518 | 16 | 15878 | 2048 | 244 | 1 |
| FeTaQA | 653.2 | 38.8 | 7326 | 1853 | 158 | 0 | 2003 | 1548 | 114 | 0 |
| FEVEROUS | 799.3 | 3.4 | 40669 | 2047 | 5 | 2052 | 4285 | 2048 | 4 | 195 |
| MultiWOZ | 777.2 | 154.5 | 56668 | 1656 | 196 | 0 | 7368 | 1344 | 185 | 0 |
| DART | 133.7 | 30.3 | 62659 | 406 | 258 | 0 | 5097 | 261 | 109 | 0 |
| Logic2Text | 166.1 | 26.9 | 8566 | 358 | 67 | 0 | 1092 | 347 | 60 | 0 |
| MTOP | 961.0 | 34.4 | 15667 | 1002 | 215 | 0 | 4386 | 990 | 113 | 0 |
| SlimOrca | 278.9 | 152.4 | 512069 | 2047 | 1808 | 0 | - | - | - | - |
| BIRD | 439.8 | 63.3 | 9428 | 1992 | 347 | 99 | 1534 | 1214 | 386 | 0 |
| CoSQL | 287.4 | 34.9 | 9502 | 1640 | 226 | 0 | 1300 | 535 | 190 | 0 |
| SQA | 656.9 | 34.9 | 12275 | 1812 | 1012 | 2 | 3011 | 1725 | 769 | 0 |
| Infotabs | 276.9 | 3.7 | 16538 | 1009 | 5 | 0 | 5400 | 1105 | 4 | 0 |
| WikiTableText | 149.6 | 27.4 | 10000 | 313 | 97 | 0 | 2000 | 226 | 89 | 0 |
| Finqa | 1230.3 | 21.0 | 6251 | 2040 | 72 | 186 | 1147 | 2048 | 61 | 25 |

Table 1: Token sequence length statistics for each dataset in our train and test sets. Input and output statistics are in tokens. We report the number of examples which have been truncated in each dataset.

context window with rare truncations. We discard examples for which even this structured input truncation is insufficient (e.g. the output is too long). During inference, we set the input token length to 2048, to allow even more structured information to be placed within the input context. We set the maximum generation length to 1024, which is sufficient for all correct responses in all datasets. For each model, including our single-task finetuned models, we choose the best performing checkpoint of the 3-epoch checkpoints.

## 4 Experiments

**Baselines**   Firstly, to illustrate the current performance of language models on SKG tasks, we evaluate ChatGPT (GPT-3.5-turbo) and the base model CodeLlama-7B-Instruct under a 1-shot setting. Our prompting scheme, using the same linearized knowledge structures as in our held-in training, sees them struggle across the board with many tasks due to the unseen structure knowledge format. Although ChatGPT is superior on text-based tasks, its performance is lackluster on SKG tasks. Its gap with SoTA models is as significant as 35%.

**Held-in Results**   To evaluate the benefits of our instruction-tuning dataset mix, we also run single-task baseline (each a 7B model) on each task and report their individual performance. We again use CodeLlama-7B-Instruct as the base model for each, and match each single task model on the same number of epochs (3) that was used to train the multitask models, ensuring that each model has seen the same data the same number of times. We observe that our multi-task models outperform these single-task models on nearly every task, with some by a considerable margin of up to 7%. This demonstrates the effectiveness of our instruction tuning dataset and supports the presence of cross-task generalization.

When compared to the 18 task-specific USKG models, `StructLM`-7B can surpass USKG by a average of 2%. From a parameter-count perspective, each of the USKG models is a T5-3B model, which means over the entire held-in set, these results require 54B parameters. Our 7B-M `StructLM` in comparison can be viewed as being nearly 8x as parameter efficient while

| Dataset | Metric | SoTA | ChatGPT | Base-M | Base | ST | FLAN-UL2 | TableLlama | USKG | StructLM (Ours) | | | | Δ |
|---|---|---|---|---|---|---|---|---|---|---|---|---|---|---|
| Size | - | - | - | 7B | 7B | 7B×18 | 20B | 7B | 3B×18 | 7B-M | 7B | 13B | 34B | - |
| **Held In** | | | | | | | | | | | | | | |
| ToTTo | BLEU | 49.9 | 20.7 | 17.9 | 17.5 | 48.8 | - | - | 49.0 | 49.8 | 49.4 | 49.3 | 50.2 | +0.3 |
| GrailQA | EM | 77.1 | 9.3 | 1.5 | 1.0 | 77.0 | - | - | 70.1 | 81.2 | 80.4 | 79.2 | 82.2 | +5.1 |
| SQL2Text | Blec | 94.8 | 88.6 | 90.7 | 82.9 | 95.2 | - | - | 94.8 | 95.2 | 93.8 | 88.5 | 92.6 | +0.4 |
| MMQA | F1 | 85.3 | 59.6 | 41.5 | 30.7 | 81.5 | - | - | 85.3 | 85.5 | 85.2 | 86.0 | 88.1 | +2.8 |
| Spider | EM | 80.5 | 43.8 | 31.0 | 5.2 | 67.3 | - | - | 71.8 | 72.4 | 72.4 | 74.1 | 74.6 | -5.9 |
| KVRet | All Micro | 67.9 | 52.9 | 34.4 | 39.5 | 70.9 | - | - | 67.9 | 72.2 | 72.6 | 69.5 | 69.3 | +4.7 |
| HybridQA | Acc | 68.4 | 23.7 | 12.9 | 2.3 | 58.4 | 61.0 | - | 59.4 | 62.6 | 59.2 | 59.1 | 61.1 | -5.8 |
| SParC | EM | 68.2 | 32.2 | 23.7 | 3.2 | 62.3 | - | - | 61.5 | 63.3 | 61.9 | 64.9 | 63.4 | -3.3 |
| CompWebQ | Acc | 76.8 | 48.9 | 30.9 | 3.1 | 75.6 | 75.9 | - | 73.3 | 79.9 | 78.3 | 80.4 | 81.9 | +5.1 |
| TabFact | Acc | 93.0 | 62.4 | 25.7 | 0.0 | 79.6 | 87.1 | 82.5 | 83.7 | 84.6 | 80.8 | 84.7 | 86.6 | -6.4 |
| WikiTQ | All Ex | 65.9 | 24.8 | 6.7 | 0.2 | 45.7 | 54.6 | - | 49.3 | 56.8 | 50.1 | 53.4 | 55.7 | -9.1 |
| WikiSQL | All Ex | 93.0 | 31.5 | 21.5 | 0.4 | 86.5 | 87.3 | - | 86.0 | 87.0 | 88.7 | 87.2 | 87.6 | -4.3 |
| FeTaQA | BLEU | 39.0 | 7.4 | 13.7 | 5.6 | 33.8 | 35.8 | 39.0 | 33.4 | 37.5 | 36.0 | 35.6 | 37.5 | -1.5 |
| FEVEROUS | Acc | 85.6 | 57.8 | 73.2 | 58.4 | 78.1 | 85.6 | - | 82.4 | 85.9 | 84.4 | 85.0 | 85.7 | +0.3 |
| MultiWOZ | Joint Acc | 60.6 | 8.9 | 0.3 | 0.0 | 53.0 | - | - | 55.4 | 55.4 | 54.5 | 53.0 | 53.8 | -5.2 |
| DART | BLEU | 52.0 | 59.0 | 47.4 | 54.6 | 60.3 | 50.4 | - | 46.7 | 63.2 | 62.2 | 61.4 | 61.8 | +11.2 |
| Logic2Text | Blec | 95.3 | 78.5 | 81.5 | 59.1 | 89.5 | - | - | 91.4 | 89.5 | 88.9 | 90.1 | 89.1 | -5.2 |
| MTOP | EM | 87.5 | 1.4 | 0.8 | 0.0 | 77.4 | 87.5 | - | 86.8 | 75.8 | 81.2 | 81.6 | 82.1 | -5.4 |
| **Average** | | 74.9 | 39.5 | 30.8 | 20.2 | 68.2 | - | - | 69.3 | 72.1 | 71.1 | 71.3 | 72.6 | -1.2 |
| **Held Out** | | | | | | | | | | | | | | |
| BIRD | Acc | 36.6* | 21.8 | 11.5 | 0.0 | 24.4* | 1.0 | 0 | 0 | 22.8 | 22.3 | 22.8 | 24.7 | +2.9 |
| CoSQL | EM | 58.3* | 33.7 | 26.5 | 0.2 | 52.4* | 5.1 | 0 | 0 | 52.8 | 49.8 | 52.2 | 55.0 | +21.3 |
| SQA | Acc | 70.5* | 18.7 | 7.4 | 2.3 | 60.4* | 70.1* | 0 | 0 | 42.6 | 49.7 | 36.1 | 44.2 | +31 |
| Infotabs | Acc | 75.6* | 46.9 | 49.1 | 40.2 | 68.7* | 70.3 | 16.2 | 0 | 47.2 | 55.3 | 58.1 | 61.8 | -8.5 |
| WikiTableText | BLEU | 33.7* | 3.8 | 3.9 | 5.7 | 39.8* | 19.4 | 3.4 | 0 | 17.1 | 8.3 | 9.3 | 8.8 | -2.3 |
| Finqa | Acc | 71.1* | 31.4 | 0.7 | 1.7 | 79.7* | 5.9 | 2.6 | 0 | 29.5 | 27.3 | 25.6 | 36.2 | +4.8 |
| **Average** | | 57.6* | 26.1 | 16.5 | 8.4 | 54.2* | 28.6* | 3.7 | 0 | 35.3 | 35.5 | 34.0 | 38.4 | +8.2 |

Table 2: The overall evaluation results of our model against other baselines. 7B-M was trained with Mistral-7B as the base model. Cells with "-" in the held-in part mean that the model did not train on this dataset, and results are not comparable. USKG models are overfit to the held-in dataset labels, and thus cannot generalize comparably. Cells in the held-out section with "*" are held-in results. SoTA results are copied from the original papers for reference. ST refers to the single-task fine-tuning result of CodeLlama-Instruct-7B on each dataset. BASE and BASE-M refer to the 1-shot performance of CodeLlama-Instruct-7B, and Mistral-7B-Instruct-v0.2 respectively. Δ refers to the difference between StructLM and the best known result. ⌊score⌋ denotes the state-of-the-art score on specific tasks. All StructLM held-out results are 0-shot. Specifications as to how SoTA results are selected are given in [Table 4](#).

still surpassing USKG models on 16 of 18 datasets. It is worth noting that although the single-task (ST) models are more than double the size in parameters compared to USKG, they do not perform much better on average. This fact indicates that there may be significant unused model capacity that can be better utilized via more effective training regimes, such as our instruction tuning. Regarding FLAN-UL2-20B ([Tay et al., 2023b](#)), which was also extensively trained on structure knowledge grounding tasks, StructLM outperforms it on 7 of the 9 mutually held-in datasets. Our results on held-in datasets (Tabfact and FeTaQA) are on par with TableLlama ([Zhang et al., 2023](#)), which is an LLM pre-trained on 2.6M table understanding tasks. We are vastly outperforming TableLlama on the held-out datasets.

**Held-out Results** On held out tasks, StructLM shows strong generalization performance, outperforming ChatGPT on 5 of 6 tasks. These novel tasks contain remarkably different data format than the training dataset. For example, FinQA ([Chen et al., 2021](#)) requires the model to generate a mathematical expression to answer a question in the financial domain, while BIRD ([Li et al., 2023b](#)) requires the model to interface with multiple databases. These skills are not exactly covered in the training set, thus they put high demand for the models' generalization capabilities. Such generalization capabilities are non-existent in USKG models as they are task-specific. Another table foundation model TableLlama ([Zhang et al., 2023](#)) would also fail on most of the held-out tasks with performance close to zero. In contrast, StructLM is much more capable of generalization on these novel tasks. The average improvement over TableLlama is 35%. Compared to another strong foundation

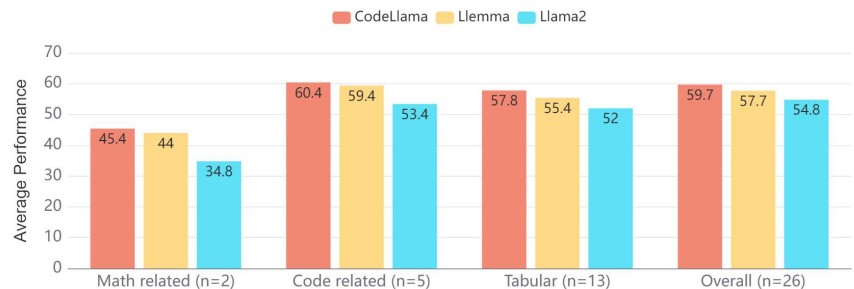

Figure 4: Effect of different pretraining curricula on SKG finetuning performance in relevant task groupings. We can observe the advantages of CodeLlma over the others.

| Purpose | Train | Eval | FT | Result |
|---|---|---|---|---|
| Schema task transfer | Spider, SParC, Logic2Text | Logic2Text | 89.47 | 89.93 |
| KT task transfer | CompWebQ, WebQSP, GrailQa, DART | DART | 60.28 | 60.34 |
| Table task transfer | FetaQA, HybridQA, WikiTQ, TabMWP, ToTTo, MMQA, WikiSQL, KVRet, Tab Fact, Feverous, Infotabs | TabFact, Feverous Infotabs | 75.46 | 80.81 |
| Summ. data type transfer | ToTTo, DART | DART | 60.28 | 61.42 |
| QA data type transfer | CompWebQ, WikiSQL | WikiSQL | 85.49 | 86.36 |

Table 3: Cross task and cross datatype transfer results. FT is an average of single-task performance over the datasets in the Eval column.

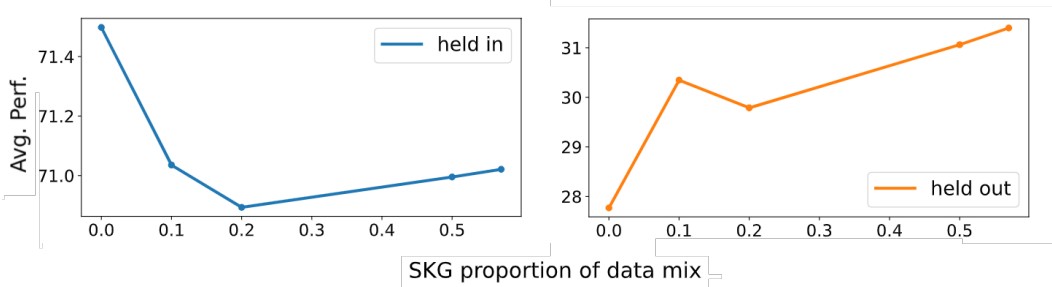

Figure 5: Effect of general instruction-following data on averaged held-out SKG dataset performance. Performance is measured as the average over evaluation metrics across all tasks within held-in or held-out groups.

structure-knowledge-grounding model Flan-UL2 20B (Tay et al., 2023b), we also outperform it on held-out tasks by an average of 10%.

Additionally, our results in Table 2 support that the Mistral base model has stronger generalization and ability to handle SKG tasks than CodeLlama, with about a 10% advantage on metrics over all tasks. We can see that this strength transfers to the fine-tuned versions of the models.

## 5 Ablation Studies

**Effect of base model pretraining data**. We ablate our choice of base model, CodeLlama-7b-Instruct, by finetuning the unspecialized Llama2-7b base model and Llemma, which is further pretrained on mathematical texts (Azerbayev et al., 2023b). Intuitively, one might guess that coding ability has the most transferability to performance on the types of SKG tasks we are studying due to the symbolic nature of programming languages and code-

writing scenarios. However, it is possible that other types of pretraining to boost reasoning ability, such as math, have even greater transferability.

Our ablation results in Table 6 can be broken down into groupings of tasks, as in Figure 4. Models pretrained on code indeed perform slightly better, and these gains are not necessarily limited to tasks which explicitly involve a grammatically regular input, or require the generation of code. Math pretraining does seem to improve the performance of the Llama2 base model, but not by as much as code pretraining. Overall, it seems that code pretraining is a useful step in training a performant model in this SKG setting, which may be due to conceptual similarity on certain tasks.

**Effect of general instruction data in mixture** As we see in Figure 5, the held-in performance is relatively unaffected by the added general examples, but held-out performance improves significantly with more general data. Empirically, we also observe that when training a large volume of task-specific input and output formats, the model becomes less capable of following instructions on new tasks in a zero-shot setting. We hypothesize that training on this general mixture helps zero-shot performance because it can reduce overfitting to the task formats in the training set.

**Cross-task and cross-format transferability** We ablate the transferability of performance between input structure knowledge types and between output task types. To test this, we train a number of tasks together and compare them to their single-task finetuned models. Our results are indicative that there is cross-task transferability of performance occurring. On tables, we see this effect clearly. This may be explained by the volume and variety of table tasks included in the training mix. In schema and knowledge triples, the performance improvement is not pronounced, perhaps due to the limited size of those datasets. Nevertheless, evidence supports that diversifying tasks on the same structured knowledge type benefits performance.

On the other hand, we see that finetuning on different datatypes with the same task (i.e. summarization) also yields benefits to performance. On the summarization and question-answering (QA) experiments, we train on both tabular and knowledge graph data. We evaluate summarization with DART and QA with WikiSQL. We see that in both cases, the extra dataset yielded about 1% improvement. Considering that the added datasets in each case organize information in a completely different way, this result suggests diversifying data types for similar tasks do indeed benefit each other as well.

## 6 Discussion

We argue that SKG is an important capability for future language models. We have seen through our experiments on ChatGPT and the Llama2 family that there is significant room for improvement. We found that we could produce a strong model by focused instruction-tuning on SKG tasks, however, we also observe that the performance difference between 7B to 34B StructLM models was not dramatic. This raises a concern about the state of SKG data: could we be approaching a performance ceiling? Combined with the fact that we were able to outperform UL2-20b, a much larger model, with our 7B model on 3 tasks, it seems that LLMs at various scales are struggling with SKG capabilities.

Indeed, grounding to structured knowledge directly in a model's input represents a challenge in reasoning and input sensitivity. However, it has a wide range of potential benefits. To meaningfully improve SKG capability, we propose that future work may explore continued pretraining of open foundation models on more structured data formats. Similar to current attempts at code or math pretraining, it is possible that pretraining models on text interleaved with tables or other types of regular data formatting will help us move towards establishing SKG as a foundational model capability.

# 7 Conclusion

In this paper, we explore the current capabilities of open language models on structured knowledge grounding tasks. We show that LLMs are currently weak at SKG tasks. To address this gap, we construct an instruction-tuning dataset mixture of 1.1M examples and release models that and achieve SOTA on 7 of 18 held-in tasks, and that outperform strong existing models such as TableLlama and Flan UL2 on held-out tasks. We also study the effects of various factors that influence the performance of our model on these task types. We hope that our work provides an updated understanding of what is achievable in the SKG domain, and can serve as a strong baseline for future improvements.

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

# A  SoTA Results

| Dataset | Metric | Split | SOTA Score | Best Performing Model |
|---|---|---|---|---|
| TabMWP | Acc | test | 94.7 | CREATOR (ChatGPT)(Qian et al., 2023) |
| ToTTo | BLEU | test | 49.9 | UniK2G (Li et al., 2024) |
| GrailQA | EM | val | 77.1 | TIARA + GAIN (T5-3B) (Shu & Yu, 2023) |
| SQL2Text | Blec | test | 94.78 | UnifiedSKG |
| MMQA | F1 | test | 85.28 | UnifiedSKG |
| Spider | EM | dev | 80.5 | RESDSQL (Li et al., 2023a) |
| KVRet | All Micro | test | 67.88 | UnifiedSKG |
| HybridQA | Acc | dev | 68.4 | S3HQA(Lee et al., 2023) |
| SParC | EM | test | 68.2 | CQR-SQL(Qi et al., 2022b) |
| CompWebQ | Acc | test | 76.8 | ChatKBQALuo et al. (2023b) |
| TabFact | Acc | test_small | 93.0 | Dater(Ye et al., 2023) |
| WikiTQ | All Ex | test | 65.9 | Dater(Ye et al., 2023) |
| WikiSQL | All Ex | test | 93.0 | SeaD+Execution-Guided Decoding(Xu et al., 2021) |
| FeTaQA | BLEU | test | 39.0 | TableLlama (Zhang et al., 2023) |
| Feverous | Acc | dev | 85.6 | FLAN UL2 20b(Tay et al., 2023b) |
| MultiWOZ | Joint Acc | test | 60.6 | TripPy + SaCLog(Dai et al., 2021) |
| Dart | BLEU | test | 52.0 | Control Prefixes (T5-large)(Clive et al., 2021) |
| Logic2Text | Blec | test | 95.3 | UnifiedSKG |
| MTOP | EM | test | 87.5 | FLAN UL2 20b(Tay et al., 2023b) |
| BIRD | Acc | dev | 36.6 | ChatGPT + COT(Li et al., 2023b) |
| CoSQL | EM | test | 58.3 | CQR-SQL(Xiao et al., 2022) |
| SQA | Acc | test | 70.5 | FLAN UL2 20b(Tay et al., 2023b) |
| Infotabs | Acc | dev | 75.6 | Infotabs paper(Gupta et al., 2020b) |
| WikiTableText | BLEU | test | 33.7 | UnifiedK2G(Li et al., 2024) |
| Finqa | Acc | private_test | 71.1 | APOLLO(Sun et al., 2022) |

Table 4: Specification of SoTA scores and their sources on the most prevalent metrics used for assessment.

Across the datasets that we evaluate, the SoTA scores are chosen based on methods that do not use agent based methods on models as large as GPT-4 for a fairer comparison. Across these datasets, we can see that many of the best performing methods are purpose-built for the type of data structure used in the task. For example, RESDSQL focuses exclusively on controlled SQL generation, DATER uses SQL-based reasoning for tabular tasks, and S3HQA focuses on table and text multi-hop QA. Compared to these methods, the performance of StructLM can be seen a strong baseline to determine if these domain-specific designs yield real benefits.

# B   SlimOrca Dataset Mixture Details

| | Metric | 0% | 10% | 20% | 50% | 57% |
|---|---|---|---|---|---|---|
| | | *Held-In Datasets* | | | | |
| TabMWP | Acc | 71.14 | 70.35 | 70.52 | 69.01 | 69.36 |
| ToTTo | BLEU | 49.78 | 49.51 | 49.47 | 49.31 | 49.38 |
| GrailQA | EM | 81.09 | 80.46 | 80.29 | 80.89 | 80.38 |
| SQL2Text | Blec | 95.07 | 94.39 | 94.49 | 94.97 | 93.81 |
| MMQA | F1 | 84.26 | 84.31 | 84.11 | 83.40 | 85.15 |
| Spider | EM | 72.92 | 71.57 | 73.40 | 72.73 | 72.44 |
| KVRet | All Micro | 71.60 | 73.90 | 70.34 | 72.25 | 72.61 |
| HybridQA | Acc | 59.23 | 59.09 | 59.09 | 59.03 | 59.17 |
| SParC | EM | 63.09 | 62.34 | 63.26 | 64.59 | 61.93 |
| CompWebQ | Acc | 80.61 | 79.15 | 78.76 | 78.73 | 78.34 |
| TabFact | Acc | 83.41 | 81.09 | 81.42 | 80.92 | 80.77 |
| WikiTQ | All Ex | 50.02 | 48.50 | 49.24 | 48.30 | 50.09 |
| WikiSQL | All Ex | 87.33 | 86.45 | 86.73 | 86.68 | 88.67 |
| FeTaQA | BLEU | 36.58 | 37.26 | 36.55 | 36.72 | 36.03 |
| Feverous | Acc | 85.02 | 84.13 | 84.11 | 83.73 | 84.41 |
| MultiWOZ | Joint Acc | 54.66 | 54.10 | 53.73 | 53.92 | 54.49 |
| Dart | BLEU | 61.38 | 61.89 | 61.08 | 62.24 | 62.24 |
| Logic2Text | Blec | 88.83 | 89.47 | 89.19 | 90.57 | 88.92 |
| MTOP | EM | 82.44 | 81.71 | 81.19 | 80.92 | 81.21 |
| | | *Held-Out Datasets* | | | | |
| BIRD | Acc | 21.30 | 22.30 | 22.30 | 23.00 | 22.30 |
| CoSQL | EM | 51.24 | 49.95 | 50.84 | 50.74 | 49.75 |
| SQA | Acc | 49.02 | 46.03 | 43.11 | 48.39 | 49.72 |
| Infotabs | Acc | 38.00 | 56.26 | 57.87 | 62.35 | 62.46 |
| WikiTableText | BLEU | 14.78 | 13.51 | 6.66 | 7.27 | 8.27 |
| Finqa | Acc | 19.70 | 24.32 | 27.55 | 25.37 | 27.29 |

Table 5: Ablation results for the mixtures of general data in the training set.

In total, we train 5 models, where the percentage represents the percent of the training data that is general. In the held out data, we see noticeable gains in generalization performance for FinQA and InfoTabs datasets. Notably, FinQA requires the generation of a python-executable math expression and InfoTabs requires an exact match to 3 previously unseen (boolean) options. WikiTableText performance seems to suffer, but is evaluated based on the BLEU score with only one target sentence. As a result, we place more emphasis on the model's 0-shot adaptation ability to new output specifications unseen in the training data.

## C  Per-dataset Pretraining Data Comparison

| Tasks | Metric | Code-LM | LLaMA | Math-LM |
|-------|--------|---------|-------|---------|
| Held-In Datasets | | | | |
| TabMWP | Acc | 71.14 | 62.96 | 66.5 |
| ToTTo | BLEU | 49.78 | 48.26 | 47.4 |
| GrailQA | EM | 81.09 | 75.72 | 77.66 |
| SQL2Text | Blec | 95.07 | 94.49 | 94.58 |
| MMQA | F1 | 84.26 | 83.96 | 82.13 |
| Spider | EM | 72.92 | 65.96 | 71.95 |
| KVRet | All Micro | 71.6 | 70.36 | 70.03 |
| HybridQA | Acc | 59.23 | 59.26 | 57.04 |
| SParC | EM | 63.09 | 56.94 | 60.35 |
| CompWebQ | Acc | 80.61 | 77.31 | 76.6 |
| TabFact | Acc | 83.41 | 80.46 | 79.47 |
| WikiTQ | All Ex | 50.02 | 45.6 | 46.89 |
| WikiSQL | All Ex | 87.33 | 83.93 | 85.49 |
| FeTaQA | BLEU | 36.58 | 34.37 | 34.1 |
| Feverous | Acc | 85.02 | 83.2 | 82.52 |
| MultiWOZ | Joint Acc | 54.66 | 55.43 | 53.79 |
| Dart | BLEU | 61.38 | 61.52 | 61.24 |
| Logic2Text | Blec | 88.83 | 88.0 | 90.38 |
| MTOP | EM | 82.44 | 77.18 | 75.56 |
| Held-Out Datasets | | | | |
| BIRD | Acc | 21.3 | 15.9 | 18.8 |
| CoSQL | EM | 51.24 | 42.8 | 48.76 |
| SQA | Acc | 49.02 | 37.03 | 49.05 |
| Infotabs | Acc | 38.0 | 4.44 | 32.54 |
| WikiTableText | BLEU | 14.78 | 13.0 | 14.82 |
| Finqa | Acc | 19.7 | 6.63 | 21.53 |

Table 6: Fine-grained evaluation results comparing finetuning done on different base models. Code refers to CodeLlama-Instruct-7B. Math refers to Llemma-7b. LLaMA refers to Llama2-7b.

In these fine-grained results we can compare the performance of Llemma, Llama, and CodeLlama in more detail. Llemma does seem to hold an advantage on tasks that involve math (TabMWP, FinQA). Math pretraining does not seem to benefit tabular tasks overall (WikiSQL, WikiTQ, HybridQA, MMQA, etc.), but does show advantages on SQL coding tasks (Spider, SparC). CodeLlama, however, seems to show a performance improvement over the base Llama model on not just coding tasks, but also math and tabular tasks as well. These findings may underscore the need to understand what constitutes "reasoning" in a language model.

## D   Prompt Format

```
[INST] <<SYS>>
You are an AI assistant that specializes in analyzing and reasoning
over structured information. You will be given a task, optionally
with some structured knowledge input. Your answer must strictly
adhere to the output format, if specified.
<</SYS>>

{instruction} {input} [/INST]
```

Figure 6: Prompt format for all SKG examples. This formatting convention follows LLama2 Touvron et al. (2023). The input contains the linearized structured data, together with any other context, question or statement. The instruction specifies the task.

# E    Held-Out Generation Examples

For illustration purposes, we provide examples of successful and unsuccessful responses of `StructLM-13B` on the FinQA held-out dataset.

### E.1    Successful `StructLM-13B` Response on FinQA

---

Use the financial data in the following passage and table to calculate the answer the question. table:

performance graph the following graph compares the yearly change in the cumulative total stockholder return for our last five full fiscal years , based upon the market price of our common stock , with the cumulative total return on a nasdaq composite index ( u.s . companies ) and a peer group , the nasdaq medical equipment-sic code 3840-3849 index , which is comprised of medical equipment companies , for that period . the performance graph assumes the investment of $ 100 on march 31 , 2007 in our common stock , the nasdaq composite index ( u.s . companies ) and the peer group index , and the reinvestment of any and all dividends. .

col : |3/31/2007 |3/31/2008 |3/31/2009 |3/31/2010 |3/31/2011 |3/31/2012 row 1 : abiomed inc |100 |96.19 |35.87 |75.55 |106.37 |162.45 row 2 : nasdaq composite index |100 |94.11 |63.12 |99.02 |114.84 |127.66 row 3 : nasdaq medical equipment sic code 3840- |100 |82.91 |41.56 |77.93 |94.54 |74.40

this graph is not 201csoliciting material 201d under regulation 14a or 14c of the rules promulgated under the securities exchange act of 1934 , is not deemed filed with the securities and exchange commission and is not to be incorporated by reference in any of our filings under the securities act of 1933 , as amended , or the exchange act whether made before or after the date hereof and irrespective of any general incorporation language in any such filing . transfer agent american stock transfer & trust company , 59 maiden lane , new york , ny 10038 , is our stock transfer agent. .

question:

what is the roi of an investment in nasdaq composite index from march 2007 to march 2010? write down a short python expression.

---

(99.02 - 100) / 100

---

In the above example, `StructLM-13B` identified the relevant quantity within the given linearized table, then determined how to calculate the required ROI quantity in a python-executable expression.

## E.2   Unsuccessful `StructLM-13B` Response on FinQA

---

Use the financial data in the following passage and table to calculate the answer the question. table:

performance graph the following graph and table compares the cumulative five-year total return provided to shareholders on our class a common stock relative to the cumulative total returns of the s&p 500 index and our customized peer group . the peer group includes cboe holdings , inc. , intercontinentalexchange group , inc . and nasdaq , inc . an investment of $ 100 ( with reinvestment of all dividends ) is assumed to have been made in our class a common stock , in the peer group and the s&p 500 index on december 31 , 2012 , and its relative performance is tracked through december 31 , 2017 .  comparison of 5 year cumulative total return* among cme group inc. , the s&p 500 index , and a peer group 12/12 12/13 12/14 12/15 12/16 cme group inc . s&p 500 peer group * $ 100 invested on 12/31/12 in stock or index , including reinvestment of dividends . fiscal year ending december 31 . copyright a9 2018 standard & poor 2019s , a division of s&p global . all rights reserved . the stock price performance included in this graph is not necessarily indicative of future stock price performance. .

col : |2013 |2014 |2015 |2016 |2017 row 1 : cme group inc . |$ 164.01 |$ 194.06 |$ 208.95 |$ 279.85 |$ 370.32 row 2 : s&p 500 |132.39 |150.51 |152.59 |170.84 |208.14 row 3 : peer group |176.61 |187.48 |219.99 |249.31 |323.23

unregistered sales of equity securities during the past three years there have not been any unregistered sales by the company of equity securities. .

question:

in 2017 what was the ratio of the the cme group inc . stock perfomamce to the s&p write down a short python expression.

---

(cme_group_inc_stock_performance / s&p_500_stock_performance)

---

In the above example, `StructLM-13B` understood how to calculate the required quantity, but could not select the relevant numbers from the given table.

## Limitations

The collection process used to construct the training data for StructLM tries to include a wide a variety of data types. As we have seen, there is evidence that this diversity is capable of affording transferable benefits to each dataset in the mixture. However, the tasks that we train and evaluate on are still academic datasets which have each been curated and designed for a specific purpose. While these are indeed diverse, the SKG domain relies on specific formatting and prompting conventions, which may result in our models having unnecessary specificity towards the conventions within our train set. To develop a clearer picture of how SKG performs as its own domain, we may require larger scale datasets with more heterogeneous formatting conventions. Further opportunities for training more robust SKG models may lie in increasing the diversity of structured data types in this way.

Additionally, while we have tried to evaluate our models to the best of our ability, many of the tasks of our held-out datasets measure accuracy through a heuristic matching step of a model's output. In zero or few-shot settings, it is quite challenging to exactly control the generations of an autoregressive transformer to be adherent to a certain rule or grammar, and this has been a subject of study in may of the other works cited in Table 4. We note that because of this reality, poor results in zero or few-shot context may betray the existence of useful representations that the model has already learned. Without further prompting or finetuning efforts, it may be difficult to bring these capabilities to light. As such, another opportunity for improvement upon our methods may involve more flexible constrained methods of language model evaluation.

