# OpenReview forum: "StructLM: Towards Building Generalist Models for Structured Knowledge Grounding"
_colmweb.org/COLM/2024/Conference — COLM_

### Official Review · Reviewer_fFrk · 2024-05-09

**Rating:** 7
**Confidence:** 4
**Ethics Flag:** 1

**Summary:**

This paper presents a large SKG instruction-tuning dataset which contains 1.1 million data samples. By fine-tuning on both the structured data and general instruction data with several tasks, the derived StructLM outperforms other LMs on structured knowledge related tasks. Besides, it also exhibits generalization capability which could be transferred to unseen structure data.

**Questions To Authors:**

1. Why don't we use a RAG when using LM for structured data? What's the advantage of StructLM than RAG?

**Reasons To Accept:**

1. The proposed dataset is useful for future researchers since it contains enough structured data which is scarce in previous datasets.
2. The StructLM outperforms other LM baselines by a large margin, which demonstrates the dataset's practicality and provides a useful LM for structure data prompting.

**Reasons To Reject:**

1. The authors seems not testing StructLM on general-sense datasets, which is crutial for a LM.
2. The performance doesn't surpass the SoTA.

---

> ### Author Rebuttal · Authors · 2024-05-30
>
> Thanks for reading our work.
>
> > Why don't we use a RAG when using LM for structured data? What's the advantage of StructLM than RAG?
>
> StructLM and RAG are not mutually exclusive. StructLM is actually an important part of RAG.
> RAG is retriever + augmented generation. StructLM focuses on the "augmented generation", where the augmented knowledge is presented in structured format.
>
> On some of our evaluation tasks, we already used retrievers to narrow down to the subgraph and then feed the subgraph as augmented knowledge to StructLM.

---

### Official Review · Reviewer_3aLT · 2024-05-10

**Rating:** 7
**Confidence:** 4
**Ethics Flag:** 1

**Summary:**

*Post-rebuttal update*: Based on all the responses, my score remains unchanged. Code/data is already released which is a plus.

The is a good empirical study of LLM performance on tasks involving structured data (such as knowledge graphs, databases, tables, etc.). The idea is somewhat simple -- measure Structured Knowledge Grounding capabilities analogous to how performance is measured on math related or other knowledge tasks. Nevertheless similar benchmarks don't exist in the literature.

**Reasons To Accept:**

* The paper is well written, clear and has all the necessary "ingredients".
* A well evaluated and formulated empirical study
* Substantial experiments on several different tasks and datasets. I also like that the authors bring in related tasks such as fact verification into the scope of knowledge grounding. This makes a lot of sense but not enough attention has been paid to this task in the LLM evaluation literature.
* Interesting findings and a fresh view of combining LLMs with structured knowledge.

**Reasons To Reject:**

* Code is not shared as part of the submission but I assume it would be released?

---

> ### Author Rebuttal · Authors · 2024-05-30
>
> Thanks for reading our work, and your kind words on the paper and the value of our contributions. We have already released the code and datasets but unfortunately have not anonymized them so that they can be attached here.

---

### Official Review · Reviewer_BN8n · 2024-05-12

**Rating:** 7
**Confidence:** 4
**Ethics Flag:** 1

**Summary:**

This work explores the problem of how large language models (LLMs) handle structured data such as that found in tables, databases, and knowledge graphs. This issue is crucial because, as Jiang et al. (2023) have shown, current powerful LLMs struggle with structured data.

The paper develops a vast and diverse dataset for instruction tuning, comprising over 1 million samples, and trains various models, termed StructLM, ranging from 7B to 34B parameters. Importantly, these models are general-purpose and are not fine-tuned on specific sources of structured data, yet they perform quite strongly on novel tasks.

Additionally, the use of code-trained LLMs is an interesting exploration. The insight that code training is beneficial for structured data tasks can be valuable for others, although this insight has already been recognized as the inclusion of code in LLM pre-training data seems to improve reasoning capabilities in LLMs.

The evaluation is thorough, and appropriate baselines are compared. Overall, this paper appears to be a good fit for COLM, contributing to the community by releasing the model weights, training dataset, and relevant code, which encourages further research and application in structured knowledge grounding (SKG).

**Questions To Authors:**

Increasing the model size from 7B to 34B parameters results in only marginal improvements. Why do the authors think this is the case with reference to scaling laws? Is more data required, or is something else at play here?

**Reasons To Accept:**

Overall, this paper appears to be a good fit for COLM, contributing to the community by releasing the model weights, training dataset, and relevant code, which encourages further research and application in structured knowledge grounding (SKG).

**Reasons To Reject:**

There are not many conceptually novel ideas, and the paper is largely a model training effort, but this is not a show-stopper, and is indeed common and valuable given the weights and dataset that are contributed to the community.

---

> ### Author Rebuttal · Authors · 2024-05-30
>
> Thank you for reading our work and your positive outlook on the value of our contributions.
>
> > Increasing the model size from 7B to 34B parameters results in only marginal improvements. Why do the authors think this is the case with reference to scaling laws? Is more data required, or is something else at play here?
>
> We do observe that the model performance improves as the parameter size increases. However, the curve is more flat than on other coding and math tasks. The problem is in the existing pre-training corpus, which largely omits the structured text (like tables, graphs) on the web. This leads to underfitting in structure text data. Therefore, our argument is that: with the current pre-training data, simply enlarging the model size cannot lead to significant improvement. We advocate adding more structured web data to the pre-training corpus to help LLMs achieve better performance on SKG tasks.

---

### Official Review · Reviewer_dWUP · 2024-05-13

**Rating:** 8
**Confidence:** 4
**Ethics Flag:** 1

**Summary:**

The paper develops a 1,1Ml example dataset to finetune LLMs for Structured Knowledge Grounding. Extensive experimentation shows that the SKG-specific finetuning effectively improves the sota on a few benchmarks, indicating that SKG may have been neglected in current instruction-tuned models; the study also identifies interesting limitations.

**Reasons To Accept:**

The topic is very important. Reasoning with tables is Achilles’s heel of many strong mainstream models. The authors state that they will release the dataset; this could be a valuable resource for the community. Experimentation is extensive; the paper is well written. Interesting discussion and limitation sections.

**Reasons To Reject:**

The technological or scientific novelty is somehow limited.  information about how the dataset was created is not very specific

---

> ### Author Rebuttal · Authors · 2024-05-30
>
> Thank you very much for reading our work and your positive outlook on the value of our contributions. We have released all our models and datasets already, and this should make the concerns about the dataset details more transparent.

---

### Official Review · Reviewer_qYA4 · 2024-05-23

**Rating:** 6
**Confidence:** 4
**Ethics Flag:** 1

**Summary:**

This work introduces StructLM, which provides a series of unified fine-tuned language models (ranging from 7B, 13B and 34B) aiming to solve structured knowledge grounding tasks. The SFT data are collected from 19 existing hold-in SKG tasks (whether 18 or 19 is not sure, typos in Section 3.1), and general instruction tuning corpus. In total, the collected data size is 1.1 M. Experiments demonstrate the effectiveness of StructLM on 18 tasks, outperforming UnifiedSKG on 11 of them using 7B model and 14 of them with 34B model. And they achieve SOTA on 7 tasks among them.

**Questions To Authors:**

1. Section 3.1: "We report results on 18 held-in and 6 held-out tasks". It seems the actual number of held-in tasks/datasets is 19, not 18, according to Table 1, which gives 19+6=25 SKG tasks.

2. Will the collected and preprocessed 1.1 M data be open-sourced? This may promote the development of further SFT work on SKG tasks in future.

**Reasons To Accept:**

1. This work presents abundant experiements on many SKG tasks to demonstrate the effectiveness of SFT on LLMs in solving SKG tasks.

2. The results are satisfactory and achieve SOTA on 7 tasks, while beating other prompting method and similar unified multi-tasking model competitors.

**Reasons To Reject:**

1. Although the author emphasizes that StructLM outperforms UnifiedSKG, it is worth mentioning that the largest model of UnifiedSKG is 3B, less than one half of the smallest StructLM. This gives rather unfair comparison.

2. From the engineering perspective, this work is self-contained and complete. However, from the perspective of academics, this work makes little contribution, which merely collects abundant SKG data to train a larger model. And all SKG data are collected from existing benchmarks. This further discounts the effort of this work.

---

> ### Author Rebuttal · Authors · 2024-05-30
>
> > Section 3.1: "We report results on 18 held-in and 6 held-out tasks". It seems the actual number of held-in tasks/datasets is 19, not 18, according to Table 1, which gives 19+6=25 SKG tasks.
>
> It is correct that we have 19 tasks in our held-in dataset, however, we omit reporting TabMWP results in the main table (Table 2) because we could not find baselines that also have TabMWP as a held-in dataset during finetuning. Nevertheless, below we are providing the results of TabMWP for StructLM models. Please feel free to compare with the other results on the leaderboard here.
>
> | Metric | SOTA (Chameleon GPT-4) | StructLM-Mistral-7b | StructLM-7b | StructLM-13b | StructLM-34b |
> | --- | --- | --- | --- | --- | --- |
> | Accuracy | 98.78 | 74.6 | 69.4 | 71.2 | 75.8 |
>
> > Will the collected and preprocessed 1.1 M data be open-sourced? This may promote the development of further SFT work on SKG tasks in future.
>
> yes, the data has been open sourced.
>
> Thanks for reading our work and giving your feedback. Our intention with this work was to provide an updated strong baseline for SKG tasks and transparently help the community track progress of the possible state of the art on this important task category. We hope that the dataset we construct and the models that we have trained can have value in this regard.

---

> > ### Comment · Reviewer_qYA4 · 2024-06-03
> > **Revision of scores**
> >
> > I have read the response and willing to improve the score to 6: Marginally above acceptance threshold. The reason is that:
> >
> > 1.  The author explains the reason why he/she misses one entry in the tasks table, which seems reasonable.
> >
> > 2. The author promises that their preprocessed data will be open-sourced which will greatly promote the development of community on table tasks.
> >
> > 3. However, I still maintain my opinion that the novelty of this work is eaten by previous work UnifiedSKG, and the author offers an unfair comparison regarding model size with that work.

---

### Decision · Program_Chairs · 2024-07-10

**Decision:**

Accept

**Comment:**

Summary:
The paper introduces "StructLM," a series of fine-tuned language models aimed at improving performance on Structured Knowledge Grounding (SKG) tasks. The models based on the CodeLlama, ranging from 7B to 34B parameters, are trained on a dataset comprising 1.1 million examples collected from existing SKG tasks and general instruction tuning corpus. The paper demonstrates that StructLM outperforms the current state-of-the-art (SOTA) on several SKG benchmarks.

Strengths:
- The paper presents a thorough evaluation of StructLM across multiple SKG tasks, showing substantial improvements over existing models, including achieving SOTA on several benchmarks.
- The introduction of a large, diverse dataset for instruction tuning is a valuable contribution to the field. The authors also commit to releasing the dataset, model weights, and code, which will benefit the broader research community.
- The paper is well-written and clearly presents its methods, findings, and implications. The empirical study is well-formulated, and the integration of related tasks like fact verification into SKG is a noteworthy approach.

Weaknesses:
- As pointed out by multiple reviewers, the paper's primary contribution lies in the engineering and dataset collection rather than introducing novel scientific concepts. The approach of using code-trained LLMs for structured data tasks, while beneficial, is not entirely new.
- Also, some reviewers mentioned that the comparison between StructLM and other prior models such as UnifiedSKG may be considered unfair due to the significant difference in model sizes. The largest UnifiedSKG model is 3B, whereas the smallest StructLM model is 7B.

Overall, the paper makes a solid contribution to the field of structured knowledge grounding by providing a comprehensive dataset and fine-tuned models that achieve significant improvements over existing approaches. The paper is well-structured and offers practical resources for future research, which enhances its value to the community.